# Structures reveal gatekeeping of the mitochondrial Ca$^{2+}$ uniporter by MICU1-MICU2

Chongyuan Wang[1], Agata Jacewicz[1], Bryce D Delgado[1,2], Rozbeh Baradaran[1], Stephen Barstow Long[1]*

[1]Structural Biology Program, Memorial Sloan Kettering Cancer Center, New York, United States; [2]Graduate Program in Biochemistry and Structural Biology, Cell and Developmental Biology, and Molecular Biology, Weill Cornell Medicine Graduate School of Medical Sciences, New York, United States

**Abstract** The mitochondrial calcium uniporter is a Ca$^{2+}$-gated ion channel complex that controls mitochondrial Ca$^{2+}$ entry and regulates cell metabolism. MCU and EMRE form the channel while Ca$^{2+}$-dependent regulation is conferred by MICU1 and MICU2 through an enigmatic process. We present a cryo-EM structure of an MCU-EMRE-MICU1-MICU2 holocomplex comprising MCU and EMRE subunits from the beetle Tribolium castaneum in complex with a human MICU1-MICU2 heterodimer at 3.3 Å resolution. With analogy to how neuronal channels are blocked by protein toxins, a uniporter interaction domain on MICU1 binds to a channel receptor site comprising MCU and EMRE subunits to inhibit ion flow under resting Ca$^{2+}$ conditions. A Ca$^{2+}$-bound structure of MICU1-MICU2 at 3.1 Å resolution indicates how Ca$^{2+}$-dependent changes enable dynamic response to cytosolic Ca$^{2+}$ signals.

*For correspondence: longs@mskcc.org

Competing interests: The authors declare that no competing interests exist.

## Introduction

The mitochondrial calcium uniporter is a highly selective multi-subunit Ca$^{2+}$ channel of the inner mitochondrial membrane that serves as the major conduit for uptake of Ca$^{2+}$ by mitochondria and controls ATP production in response to cytosolic Ca$^{2+}$ signals (*Gunter and Pfeiffer, 1990*; *Kamer and Mootha, 2015*; *Pan et al., 2013*; *Pathak and Trebak, 2018*). The uniporter is one of the most selective Ca$^{2+}$ channels known and it is only activated once cytosolic Ca$^{2+}$ levels reach a threshold (of approximately 0.4–3 μM) (*Csordás et al., 2013*; *Kirichok et al., 2004*; *Liu et al., 2016*; *Mallilankaraman et al., 2012b*; *Payne et al., 2017*). These properties, Ca$^{2+}$ selectivity and block of ion permeation under resting conditions, are fundamentally important for preventing mitochondrial Ca$^{2+}$ overload and for maintaining the electromotive force used for ATP production because potassium and other cations outnumber Ca$^{2+}$ by more than one million to one in the cytosol. The protein MCU constitutes the pore of the uniporter through which Ca$^{2+}$ ions flow and EMRE, a small transmembrane protein that associates with MCU, is necessary for Ca$^{2+}$ uptake in metazoan organisms (*Baughman et al., 2011*; *Chaudhuri et al., 2013*; *De Stefani et al., 2011*; *Kovács-Bogdán et al., 2014*; *Sancak et al., 2013*). Ca$^{2+}$-dependent control is conferred by proteins that are unique to the uniporter – heterodimers of MICU1-MICU2 in most cells and of MICU1-MICU3 in neurons (*Ashrafi et al., 2019*; *Kamer et al., 2017*; *Kamer and Mootha, 2014*; *Patron et al., 2014*; *Patron et al., 2019*; *Payne et al., 2017*; *Perocchi et al., 2010*; *Plovanich et al., 2013*). The activity of the uniporter is further tuned by interactions with MCUR1 (*Chaudhuri et al., 2016*; *Mallilankaraman et al., 2012a*; *Vais et al., 2015*) and the MCU-like protein MCUb (*Raffaello et al., 2013*). The molecular mechanism by which the MICU1-MICU2 and MICU1-MICU3 heterodimers regulate the channel is a matter of intense study (*Csordás et al., 2013*; *Hoffman et al., 2013*;

*Kamer et al., 2017*; *Kamer and Mootha, 2014*; *Kamer et al., 2018*; *Paillard et al., 2018*; *Patron et al., 2014*; *Patron et al., 2019*; *Payne et al., 2017*; *Perocchi et al., 2010*; *Phillips et al., 2019*; *Plovanich et al., 2013*). Multiple models for this regulation have been proposed but issues as fundamental as the stoichiometry between the heterodimers and the channel remain unclear.

Studies have resolved that MICU1-MICU2 heterodimers reside in the intermembrane space (IMS) where they regulate the uniporter by responding to $[Ca^{2+}]_{IMS}$ (*Kamer and Mootha, 2014*; *Patron et al., 2014*; *Plovanich et al., 2013*). $[Ca^{2+}]_{IMS}$, which closely follows the cytosolic $Ca^{2+}$ concentration due to the permeability of the outer mitochondrial membrane, is sensed by EF-hand motifs in each protein. At the ~100 nM resting level of $[Ca^{2+}]_{cytosol}$, MICU1-MICU2 and MICU1-MICU3 heterodimers prevent ion conduction through the channel, and they permit it when $Ca^{2+}$ levels rise (*Csordás et al., 2013*; *Kamer et al., 2017*; *Kamer and Mootha, 2014*; *Liu et al., 2016*; *Mallilankaraman et al., 2012b*; *Paillard et al., 2017*; *Patron et al., 2014*; *Patron et al., 2019*; *Payne et al., 2017*; *Phillips et al., 2019*). While structures of MICU1-3 and MCU-EMRE components have been determined (*Kamer et al., 2019*; *Park et al., 2020*; *Wang et al., 2020*; *Wang et al., 2014*; *Wang et al., 2019*; *Wu et al., 2019*; *Xing et al., 2019*), it is unclear how MICU proteins interact with the channel and the mechanism by which they confer $Ca^{2+}$-dependent control to the channel is enigmatic. Further it is not known why MICU1, in particular, is necessary among MICU1-3 for regulating the channel (*Kamer and Mootha, 2014*; *Patron et al., 2014*; *Patron et al., 2019*; *Payne et al., 2017*; *Xing et al., 2019*). Here, we present a 3.3 Å resolution cryo-EM structure of the MCU-EMRE-MICU1-MICU2 assembly, hereafter referred to as the holocomplex, under resting $[Ca^{2+}]$ conditions. Together with functional data and a structure of a $Ca^{2+}$-bound MICU1-MICU2 complex, the work reveals a molecular basis for $Ca^{2+}$-dependent control of the uniporter and the unique role of MICU1 in this process.

## Results

### Structure determination of the MCU-EMRE-MICU1-MICU2 holocomplex

The ability of human MICU1 to bind MCU channels from lower-eukaryote organisms (*Phillips et al., 2019*) and our ability to obtain high-resolution structures of the metazoan MCU-EMRE channel complex from the red flour beetle, Tribolium castaneum, (*Tc*MCU-EMRE) (*Wang et al., 2020*) motivated us to pursue a cryo-EM structure of *Tc*MCU-EMRE in complex with human MICU1-MICU2. The amino acids facing the IMS, where MICU1-MICU2 resides, are identical between human and beetle MCU/EMRE and adopt analogous conformations in structures of the human and beetle channels (*Figure 1—figure supplement 1*, *Figure 1—figure supplement 2*).

A holocomplex sample that was suitable for structure determination required meticulous formulation. An MCU-EMRE subcomplex was prepared by coexpressing *Tc*MCU and *Tc*EMRE in mammalian cells, purifying the *Tc*MCU-EMRE assembly, and reconstituting it into lipid nanodiscs that include the mitochondrial lipid cardiolipin. Cardiolipin was included because we recently identified that this lipid is important for the function of the uniporter (*Wang et al., 2020*). We previously observed that a symmetry mismatch between the fourfold symmetric IMS surface of the channel and the asymmetric N-terminal domain (NTDs) in metazoan MCU channels can be a hindrance for high-resolution cryo-EM studies (*Baradaran et al., 2018*; *Wang et al., 2020*). Because the NTD is dispensable for mitochondrial $Ca^{2+}$ uptake (*Baradaran et al., 2018*; *Oxenoid et al., 2016*; *Wang et al., 2020*; *Wang et al., 2019*), we removed this domain for high-resolution structural studies. We have recently shown that the *Tc*MCU and *Tc*EMRE constructs used for structural analysis (*Tc*MCU$_{\Delta NTD}$ and *Tc*EMRE coexpressed) catalytze $Ca^{2+}$ uptake into human mitochondria (*Wang et al., 2020*).

In cells, a disulfide bond formed between flexible C-terminal regions of MICU1 and MICU2 helps stabilize the heterodimer (*Kamer et al., 2019*; *Patron et al., 2014*; *Petrungaro et al., 2015*). To mimic this condition and discourage the previously observed tendency of purified MICU1 and MICU2 to form homo-oligomers at the high protein concentrations needed for structural studies (*Kamer et al., 2019*; *Patron et al., 2014*; *Wang et al., 2014*; *Wu et al., 2019*; *Xing et al., 2019*), we expressed MICU1 and MICU2 as a single polypeptide with a linker connecting the two subunits. A further motivation for this approach was the expectation that a MICU1-MICU2 concatemer would facilitate interpretation of the cryo-EM data, as the sample might otherwise contain mixtures of homo- and heterodimers that could be difficult to distinguish computationally. By combining the

purified MICU1-MICU2 and MCU-EMRE-nanodisc assemblies we obtained a monodisperse holocomplex sample that was used for cryo-EM analysis (*Figure 1—figure supplement 3*).

Cryo-EM data were collected using a low (<1 nM) concentration of Ca$^{2+}$ in order to investigate the resting state of the holocomplex. From a large dataset containing approximately 17 million particle images, single particle analysis indicated that most channels were associated with one MICU1-MICU2 heterodimer (*Figure 1A*, *Figure 1—figure supplement 4A*). Images containing one MICU1-MICU2 heterodimer per channel yielded a 3D reconstruction of the holocomplex at 3.3 Å resolution (*Figure 1*, *Figure 1—figure supplement 4*, *Figure 1—figure supplement 5*). Two MICU1-MICU2 complexes were observed in approximately 2% of channel assemblies (*Figure 1A*, *Figure 1—figure supplement 4A*, as assessed from 2D class averages of side views). This subset of data yielded a ~ 10 Å resolution 3D reconstruction (*Figure 1—figure supplement 4C*). In it, one MICU1-MICU2 heterodimer associates with the MCU/EMRE channel in the same manner as in the high-resolution structure. Density for the second MICU1-MICU2 heterodimer is considerably weaker and positioned to the side, next to the channel, where it might interact with the lipid nanodisc. This 3D reconstruction and the 2D class averages, which display blurring of the second heterodimer (*Figure 1—figure supplement 4A*), indicate that the second one does not have a discrete binding location. Except where noted, the discussion that follows pertains to the high-resolution structure of the holocomplex that contains one MICU1-MICU2 heterodimer. The cryo-EM density of this structure is well defined, especially within the TMD and MICU1 regions, which have the highest local resolution (~3.0 Å, *Figure 1—figure supplement 5C*). The constructed atomic model has good stereochemistry and excellent agreement with the cryo-EM density (*Figure 1—figure supplement 5*, and *Table 1*).

## Overall structure of the holocomplex

MICU1 and MICU2 associate as a heterodimer and bind to the surface of the channel at the IMS entrance of its pore (*Figure 2*). Interactions with the channel are mediated solely by MICU1; MICU2 makes no visible contacts with MCU or EMRE. As in cryo-EM and X-ray structures of fungal MCU and metazoan MCU-EMRE channel complexes without MICU1-MICU2 (*Baradaran et al., 2018*;

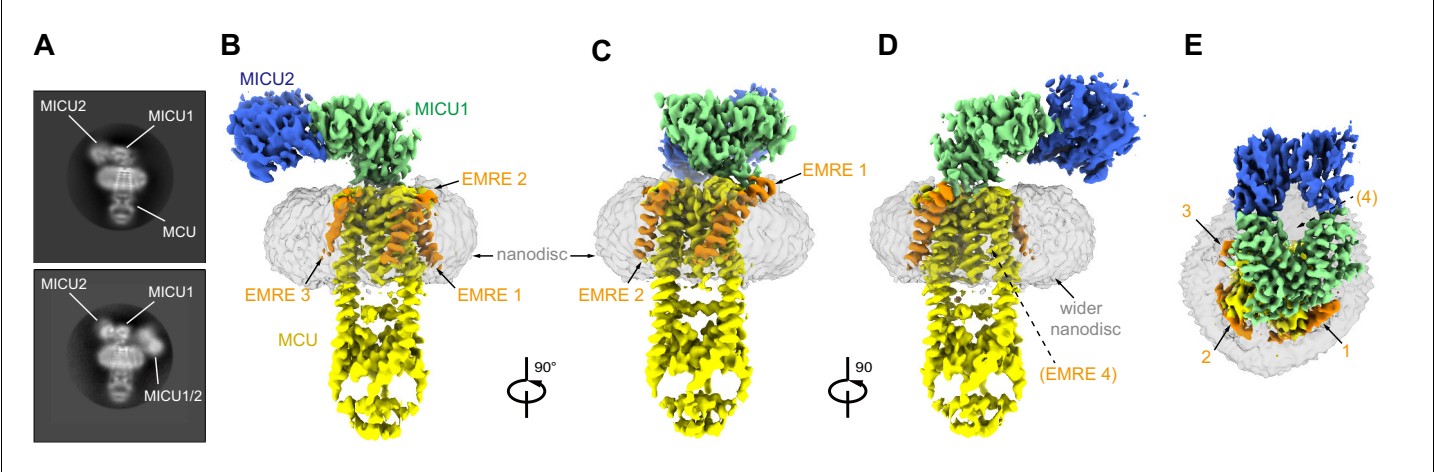

**Figure 1.** Cryo-EM reconstruction of the holocomplex. (**A**) 2D class averages showing side-views of complexes with one or two MICU1-MICU2 heterodimers associated (top and bottom panels, respectively). (**B–E**) Cryo-EM reconstruction of the holocomplex, shown in orthogonal views. Densities are colored accordingly: MCU (yellow), EMRE (orange), MICU1 (green), MICU2 (blue), and nanodisc (semitransparent gray). Numbers in (**E**) represent the locations of EMRE subunits. The location of where EMRE 4 would be expected to bind is indicated by parentheses (**D–E**).

The online version of this article includes the following figure supplement(s) for figure 1:

**Figure supplement 1.** Sequence alignments of MCU and EMRE.

**Figure supplement 2.** Comparison of the MICU1 receptor sites in human and beetle MCU-EMRE complexes.

**Figure supplement 3.** Purification of the MCU-EMRE-MICU1-MICU2 holocomplex under resting ([Ca$^{2+}$]$_{free}$ <1 nM) conditions and cryo-EM micrograph.

**Figure supplement 4.** Flowchart of cryo-EM data processing of holocomplex.

**Figure supplement 5.** Cryo-EM analysis of the holocomplex.

**Figure supplement 6.** The EF-hands of MICU1 and MICU2 are in apo conformations in the holocomplex structure.

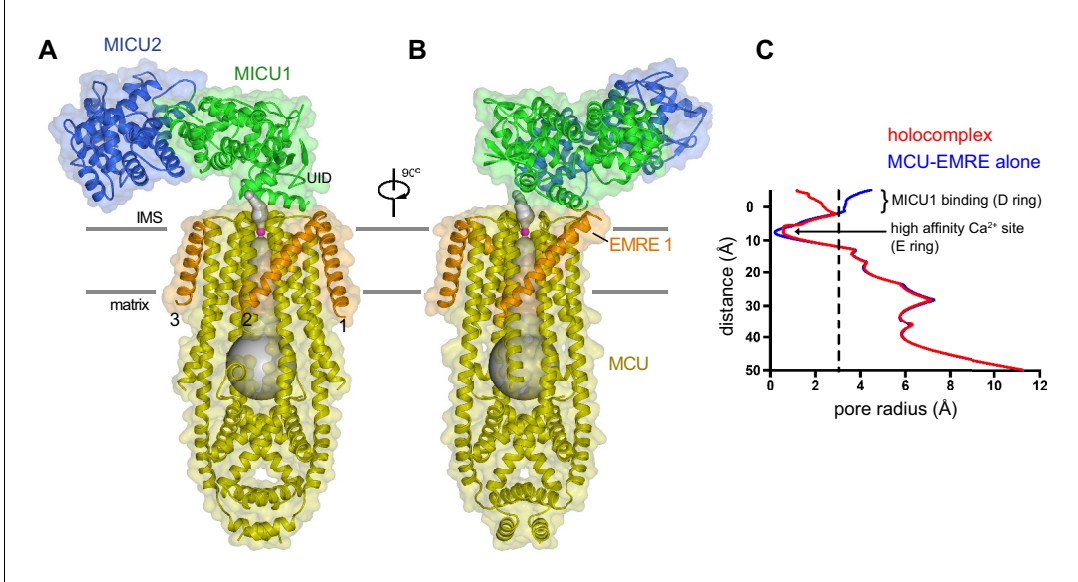

**Figure 2.** Architecture of the holocomplex. (**A and B**) Cartoon representations of the holocomplex, colored as in **Figure 1**, and shown with semitransparent surfaces. Gray bars represent approximate boundaries of the hydrophobic core of the membrane. A $Ca^{2+}$ion in the E ring is drawn as a purple sphere. The pore (semitransparent gray surface) is depicted as the minimal radial distance from its center to the nearest van der Waals protein contact. (**C**) Pore dimensions in structures of $Tc$MCU-EMRE with and without MICU1-MICU2 (red and blue, respectively). A dashed line indicates the radius of a hydrated $Ca^{2+}$ ion.

The online version of this article includes the following figure supplement(s) for figure 2:

**Figure supplement 1.** Comparison of the structures of $Tc$MCU-EMRE with and without MICU1-MICU2.

*Fan et al., 2018*; *Nguyen et al., 2018*; *Wang et al., 2020*; *Wang et al., 2019*; *Yoo et al., 2018*), the pore comprises a single ion conduction pathway through the TMD that is surrounded by four MCU subunits. EMRE subunits, located at the periphery of the TMD, consist of a single transmembrane helix flanked by short N- and C-terminal disordered regions. The conformation of the pore is indistinguishable from that observed in a cryo-EM structure of $Tc$MCU-EMRE without MICU1-MICU2 (**Figure 2C**, **Figure 2—figure supplement 1**; *Wang et al., 2020*), which suggests that unlike gating conformational changes in many other ion channels, control by MICU1-MICU2 is not accomplished through motions of pore-lining helices.

The structure of the holocomplex provides the most well-defined density to date for the ion-selectivity filter, which is formed by a fourfold assembly of the WDXXEP signature sequences of MCU subunits (**Figure 1—figure supplement 5E**). The 'D' and 'E' residues of this motif (Asp 261 and Glu 264) form two rings of acidic amino acids that line the pore of the channel and can coordinate $Ca^{2+}$ through water-mediated ('D' ring) or direct interactions ('E' ring) (*Baradaran et al., 2018*; *Fan et al., 2018*; *Nguyen et al., 2018*; *Wang et al., 2020*; *Wang et al., 2019*; *Yoo et al., 2018*). Strong density that we assign as $Ca^{2+}$ on the basis of its coordination and analogy to other structures of MCU is present within the 'E' ring (**Figure 1—figure supplement 5E**). Its presence under conditions of <1 nM $Ca^{2+}$ identifies this site as the high-affinity ($K_d \leq 2$ nM) $Ca^{2+}$ binding site previously postulated from electrical recordings made from mitoplasts that is thought to contribute to the channel's high selectivity for $Ca^{2+}$ (*Kirichok et al., 2004*). MICU1-MICU2 binding does not displace $Ca^{2+}$ from this site. The 'D' ring is directly coordinated by MICU1, and possibly because of this and/or the low concentration of $Ca^{2+}$ in the sample, density for $Ca^{2+}$ is not observed in this site. Aside from coordination of the 'D' ring, the ion conduction pathway is indistinguishable from the unliganded conformation (**Figure 2—figure supplement 1**; **Figure 2C**).

Three EMRE subunits are observed at the periphery of the TMD (EMRE 1–3). Density for a fourth, like that observed in structures without MICU1-MICU2 (*Wang et al., 2020*; *Wang et al., 2019*), is conspicuously absent (**Figure 1D and E**). MICU1 interacts with only EMRE 1, which has the most well-defined density (**Figure 1B–D**, **Figure 1—figure supplement 5D**). EMRE 2 and 3 adopt the

same conformation as EMRE 1, which is the same as the one observed without MICU1-MICU2, in which each EMRE transmembrane helix is positioned approximately 45° from the membrane normal and interacts with TM1 of an adjacent MCU subunit (*Figure 2A–B*, *Figure 2—figure supplement 1*; *Wang et al., 2020*). MICU1-MICU2 is positioned ~15 Å above where EMRE 4 would be located (*Figure 1D,E*). It is possible that interactions between EMRE 4 and MICU1-MICU2, perhaps involving the disordered acidic C-terminal tail of EMRE (*Figure 1—figure supplement 1B*), may destabilize binding of this subunit. In support of this hypothesis, nanodisc density is wider in the vicinity of where EMRE 4 would be positioned (*Figure 1D*), which suggests that it may be present within the nanodisc but displaced from the surface of MCU.

## A uniporter interaction domain and its receptor site

MICU1 contains three domains (*Wang et al., 2014*): an N-lobe and a C-lobe, which contain its $Ca^{2+}$-binding sites, and an N-terminal domain, which we name the uniporter interaction domain (UID) (*Figure 3A*). The UID mediates all observed interactions with the channel. The UID comprises three α-helices (α1, α2, α3) and a small β-sheet formed by strands β1, β2 and β3 (*Figure 3A*). Residues on α1 and α2 and a tight turn connecting these helices at Pro 124 directly contact the IMS surface of the channel (*Figure 3D,F–G*).

The binding of the UID identifies a receptor for MICU1 on the surface of the channel (*Figure 3B and C*). The receptor is relatively flat and spans three MCU subunits and one EMRE subunit. The amino acids comprising it (MCU residues: Trp 255$_B$, Tyr 258$_{A\&B}$, Ser 259$_B$, Asp 261$_{A,B\&D}$, and Ile 262$_{A\&B}$, and EMRE residue Leu 92, *Figure 3C*) are conserved among metazoan channels and are identical between human and beetle channels (*Figure 1—figure supplement 1A–B*, *Figure 1—figure supplement 2*). Apart from the absence of EMRE 4, the conformation of the receptor site is indistinguishable from the unliganded conformations of both beetle and human MCU-EMRE complexes (*Figure 1—figure supplement 2*; *Wang et al., 2020*; *Wang et al., 2019*).

The α1 helix, which is the N-terminal end of the ordered region of MICU1, is positioned horizontally on the receptor, where it contacts one EMRE subunit (EMRE 1) and two MCU subunits through a collection of van der Waals interactions and hydrogen bonds (*Figure 3B,F–G*). The interacting residues on α1 (Phe 106, Arg 107, Lys 110, Val 111, Tyr 114, Arg 117, and Tyr 121) are identical in beetle MICU1 and conserved among animals (*Figure 3—figure supplement 1B*).

A hydrophobic interaction between Phe 106 of α1 and the IMS end of EMRE's transmembrane helix at Leu 92, which is also highly conserved (*Figure 1—figure supplement 1B*), comprises the only contact observed between MICU1 and EMRE (*Figure 3G*). Although not visible in the density, MICU1 and EMRE may also interact electrostatically – an acidic C-terminal region of EMRE and a basic region preceding α1, which are disordered, have been implicated in binding (*Hoffman et al., 2013*; *Tsai et al., 2016*) and would be in close proximity. A nearby conserved basic surface of MICU1 may be another point of contact for the acidic tail of EMRE (*Figure 3—figure supplement 2*).

The mouth of the pore is coordinated by Lys 126 and Arg 129 from α2, and Ser 122 of the α1-α2 turn through a network of hydrogen bonds that involves three of the four MCU-Asp 261 residues of the 'D' ring (*Figure 3D,F*). This confirms the previous hypothesis that the 'D' ring is involved in MICU1 binding (*Paillard et al., 2018*; *Phillips et al., 2019*). The MICU1 residues participating in the hydrogen-bonding network (Tyr 114, Ser 122, Asp 125, Lys 126, Arg 129, and Asp 169) are conserved (*Figure 3—figure supplement 3*), but none were previously predicted to mediate interactions with the channel (*Figure 3—figure supplement 4*).

To assess the function of the interacting residues and to gauge their importance and relevance for control of the full-length human MCU-EMRE channel, we evaluated mitochondrial $Ca^{2+}$ uptake when wild type and mutant MICU1 proteins were expressed in human MICU1 knockout cells. As expected, robust mitochondrial $Ca^{2+}$ uptake was observed when wild type MICU1 was expressed and uptake was suppressed by mutations within the $Ca^{2+}$-binding EF-hands of MICU1 that cause it to constitutively inhibit the channel (*Figure 3E*, *Figure 3—figure supplement 5*; *Kamer and Mootha, 2014*). Residues that were observed to interact with MCU or EMRE relieve this inhibition when mutated to alanine. This includes not only Lys 126 and Arg 129 of MICU1 that interact with the 'D' ring but also amino acids that interact with MCU and EMRE at a distance from the pore (e.g. Phe 106, Arg 117, and Tyr 121). Lys 126 seems particularly crucial – a single K126A mutation essentially

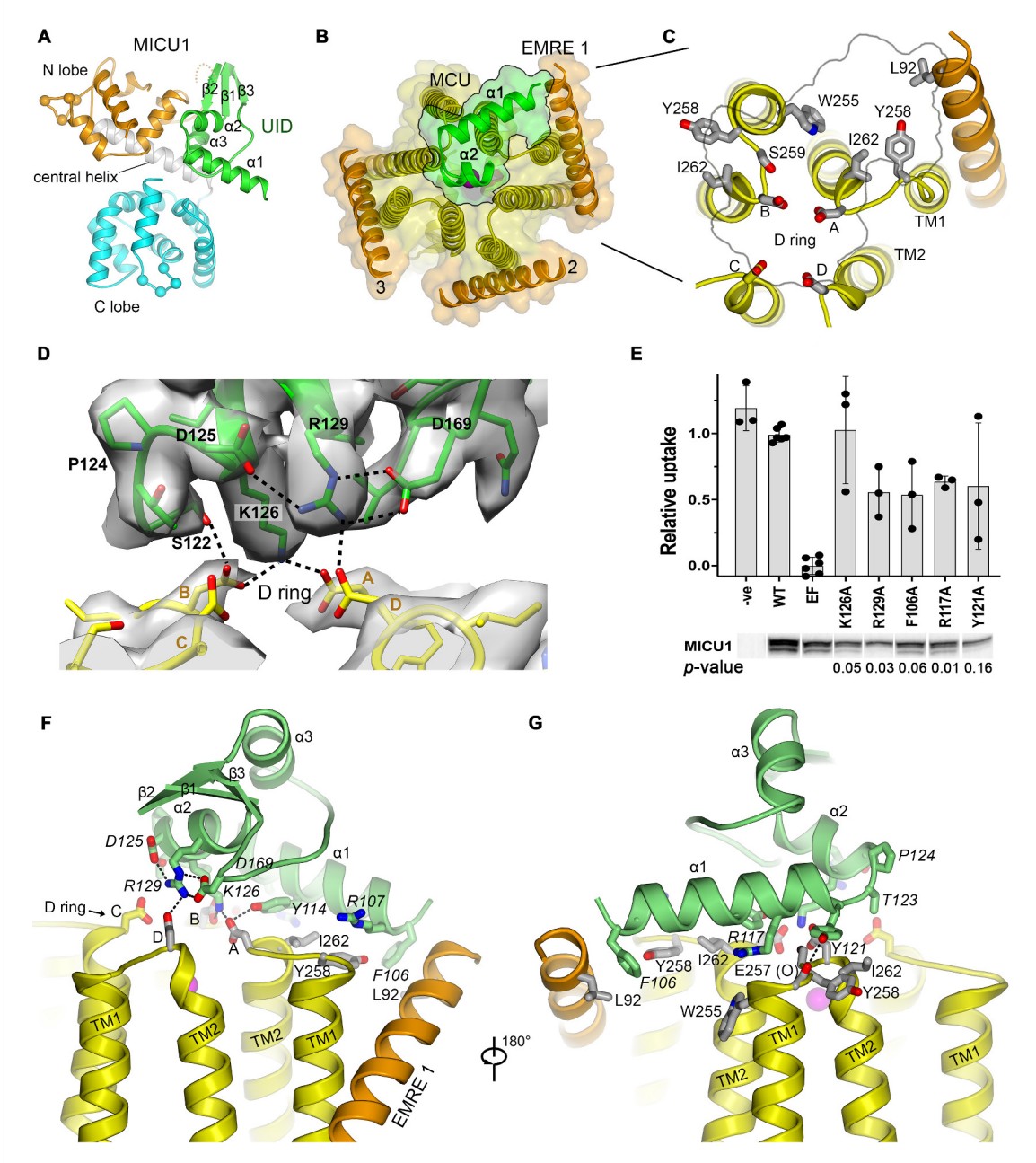

**Figure 3.** The UID and its receptor site. (A) Domain architecture of MICU1. Cartoon representation colored according to subdomains. Spheres indicate the Cα positions of amino acids that would bind Ca²⁺ in EF-hands EF1 and EF4. (B) The UID-receptor interface. A semitransparent surface-cartoon rendition of MCU and EMRE is shown, viewed from the IMS. An outline marks the boundaries of the interface of the UID with the channel. α1 and α2 of the UID are depicted as green ribbons. (C) Close up view of the receptor site, with residues that interact with the UID drawn as sticks. (D) Cryo-EM density and interactions at the IMS mouth of the pore. The interface region between MICU1 and the D ring is shown (MICU1, green; MCU, yellow), with density depicted as a semitransparent surface. Dashed lines indicate hydrogen bonds. (E) Mutagenesis of the UID and the effect on mitochondrial Ca²⁺ uptake. The indicated mutants of human MICU1, made in the background of disrupted EF-hand domains (EF), were transiently expressed in MICU1 knockout cells and uptake was quantified relative to wild type (WT) MICU1 and disrupted EF-hand (EF) controls. Western blots demonstrate expression. '-ve' indicates untransfected cells. (mean ±s.d., independent experiments: n = 6 for WT and EF controls, and n = 3 for the remainder. Student's t-Test p-values for mutants relative to the EF construct, calculated using the two-tailed distribution with unequal variance method, are listed). (F and G) Views showing UID-receptor site interactions (sticks). MICU1 residues are in italics.

The online version of this article includes the following figure supplement(s) for figure 3:

**Figure supplement 1.** The UID is unique to MICU1.

**Figure supplement 2.** Electrostatic surface of MICU1-MICU2 and possible interaction site for the C-terminal tail of EMRE.

*Figure 3 continued on next page*

eliminates inhibition by MICU1. These studies verify the structurally observed interaction and indicate its importance for control of the full-length human channel.

The regions of the UID that interact with the channel are absent in MICU2 or MICU3 (*Figure 3—figure supplement 1*). The α1 helix, the conserved Pro 124 in the bend between α1 and α2, and residues that interact with the 'D' ring are unique to MICU1. This explains why MICU1 is distinct from MICU2 and MICU3 in its ability to interact with MCU/EMRE, and it explains the observation that MICU2 and MICU3 have non-redundant roles in comparison to MICU1 in the regulation of the channel (*Kamer and Mootha, 2014*; *Patron et al., 2014*; *Patron et al., 2019*; *Payne et al., 2017*; *Xing et al., 2019*).

The interaction with the pore is reminiscent of the interaction between μ-conotoxin KIIIA, a small pore-blocking peptide toxin found in the venom of a cone snail, with the extracellular surface of the neuronal Na$_v$1.2 channel (*Pan et al., 2019*; *Zhang et al., 2007*; *Figure 3—figure supplement 6*). A short α-helical region of the toxin lies horizontally at the entrance of the pore of Na$_v$1.2, in a similar manner to the binding of the α2 helix of MICU1 with the uniporter. As in the UID-uniporter interface, an arginine and a lysine residue that are separated by one turn of the toxin's α-helix coordinate acidic amino acids at the entrance of the selectivity filter (*Figure 3D* and *Figure 3—figure supplement 6*). The binding of the toxin is slightly off-center such that two of four analogous acidic amino acids are coordinated and the pore is not completely occluded – nevertheless this toxin blocks approximately 90% of Na$^+$ current though the channel (*Zhang et al., 2007*; *Figure 3—figure supplement 6*). The UID interacts with three of the four 'D'-ring amino acids at the mouth of the uniporter's pore and the occlusion of the pore is more extensive than observed for Na$_v$1.2 by μ-conotoxin KIIIA (*Figure 2*, *Figure 3*, *Figure 3—figure supplement 6*). The structure of the holocomplex and its similarity to a pore-blocking toxin complex indicate that MICU1 binding inhibits the flow of Ca$^{2+}$ ions by both obstructing the IMS entrance of the pore and by shielding the negative charge of the 'D' ring through coordination with basic amino acids.

## MICU1-MICU2 and Ca$^{2+}$-induced changes

The MICU1-MICU2 portion of the holocomplex is roughly parallelogram shaped, with overall dimensions of approximately 70 × 65 × 35 Å, under resting conditions in which its EF-hand domains are in Ca$^{2+}$-free apo states (*Figure 4A*). One of its large relatively-flat surfaces faces and curves slightly toward the membrane due to a slight bend between MICU1 and MICU2 (*Figure 2A*). The heterodimer has pseudo twofold symmetry and two analogous interfaces between the subunits (*Figure 4A*, *Figure 4—figure supplement 1A–F*). Hydrophobic amino acids predominate the interfaces, with MICU1-Met 229, MICU1-Phe 383, MICU2-Met 183, and MICU2-Met 337 making analogous and particularly extensive contacts within them (*Figure 4—figure supplement 1A–C*). The interfacial residues are conserved in MICU3 (*Figure 4—figure supplement 2*), which suggests that the neuron-specific MICU1-MICU3 heterodimer has a similar arrangement. As would be expected under resting [Ca$^{2+}$] conditions, all four Ca$^{2+}$-binding EF-hand motifs of the MICU1-MICU2 heterodimer ('EF1' and 'EF4' of both subunits) are in the apo state and the heterodimer is superimposable with a previous apo structure of MICU1-MICU2 alone (*Park et al., 2020*; *Figure 1—figure supplement 6*, *Figure 4—figure supplement 3A*).

From biochemical analysis of the purified protein, and in acord with a previous report that MICU1-MICU2 has an affinity for cardiolipin (*Kamer et al., 2017*), we found that the heterodimer associates with lipid membranes and lipid nanodiscs that contain cardiolipin (*Figure 4—figure supplement 4*, *Figure 4F*). This association is independent of Ca$^{2+}$ - both Ca$^{2+}$-free and Ca$^{2+}$-bound heterodimers bind to liposomes. To investigate the conformational change in MICU1-MICU2 upon Ca$^{2+}$ binding, we determined a cryo-EM structure of Ca$^{2+}$-bound MICU1-MICU2 in complex with a lipid nanodisc containing cardiolipin at 3.1 Å resolution (*Figure 4F*, *Figure 4—figure supplement*

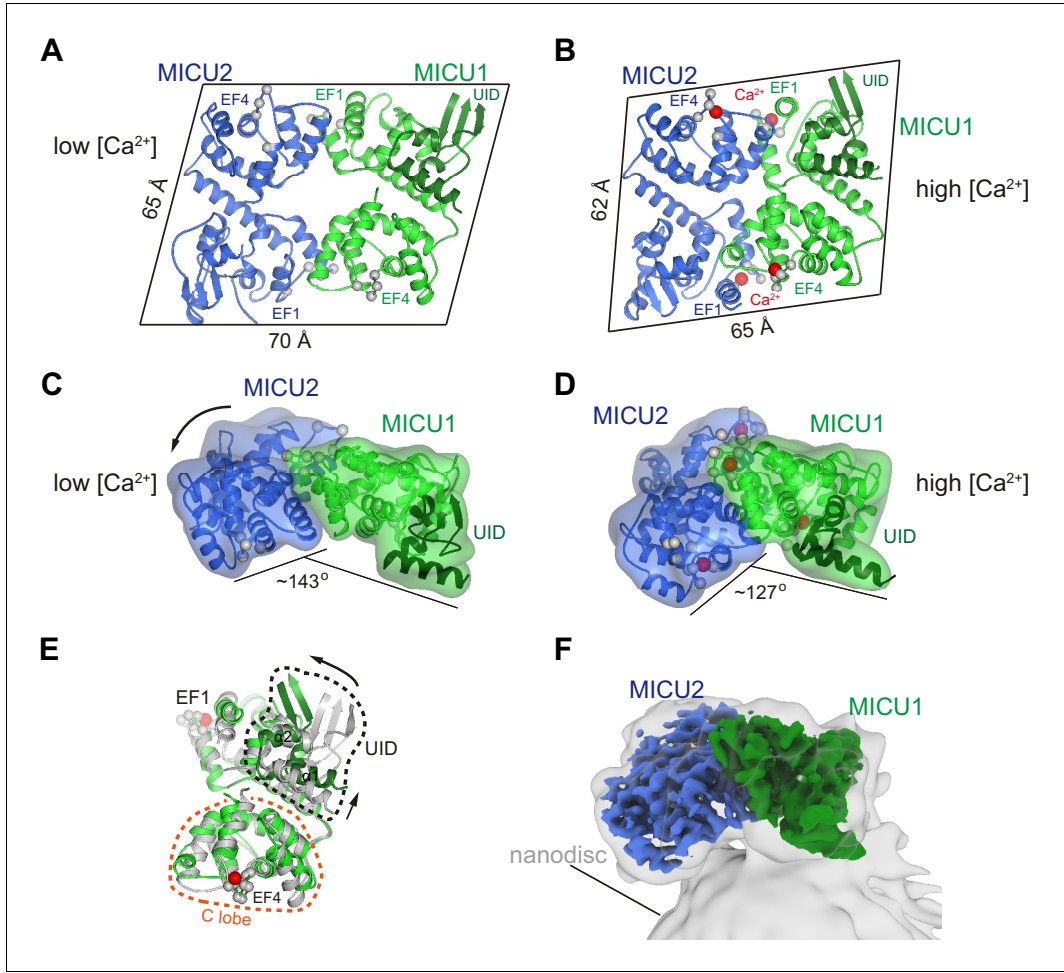

**Figure 4.** $Ca^{2+}$-dependent changes in MICU1-MICU2. (**A**) Cartoon representation of the apo MICU1-MICU2 heterodimer. Gray spheres denote EF-hand residues. (**B**) Overall structure of $Ca^{2+}$-bound MICU1-MICU2. Red spheres indicate bound $Ca^{2+}$ ions. (**C–D**) Bend between MICU1 and MICU2 induced by $Ca^{2+}$ binding (C, apo; D, $Ca^{2+}$-bound). Overall structures are shown from the side with semitransparent molecular surfaces. (**E**) Superposition of apo (gray) and $Ca^{2+}$-bound (green) MICU1 highlighting a rotation of the UID upon $Ca^{2+}$ binding. (**F**) Cryo-EM density depicting the $Ca^{2+}$-bound MICU1-MICU2 complex associated with a lipid nanodisc. The online version of this article includes the following figure supplement(s) for figure 4:

**Figure supplement 1.** MICU1-MICU2 interfaces and $Ca^{2+}$-binding.

**Figure supplement 2.** Structure based sequence alignments of MICU proteins.

**Figure supplement 3.** Comparisons of MICU1, MICU2 and MICU3 in apo and $Ca^{2+}$-bound conformations.

**Figure supplement 4.** The MICU1-MICU2 regulatory complex binds to liposomes in both $Ca^{2+}$ and $Ca^{2+}$-free conditions.

**Figure supplement 5.** Flowchart of the cryo-EM data processing of the $Ca^{2+}$-bound MICU1-MICU2 complex.

**Figure supplement 6.** Cryo-EM analysis of the $Ca^{2+}$-bound MICU1-MICU2 complex.

---

**5**, *Figure 4—figure supplement 6*, and *Table 1*). The side of the heterodimer that associates with the nanodisc is the same as the one that faces the channel (*Figures 1A* and *4F*). The structure reveals a substantial conformational change upon $Ca^{2+}$ binding that makes the heterodimer more compact and markedly more bent between MICU1 and MICU2 (*Figure 4A–D*). $Ca^{2+}$ dependent rearrangements of the EF-hand motifs that involve their helix turn helix elements produce this bend (*Figure 4C–D*, *Figure 4—figure supplement 1*). $Ca^{2+}$ binding also induces a rotation of the UID relative to MICU1 (*Figure 4E*).

We sought to understand how $Ca^{2+}$-dependent conformational changes in MICU1-MICU2 might confer $Ca^{2+}$-dependent control to the channel (*Figure 5*). Superimposing the $Ca^{2+}$-bound MICU1-

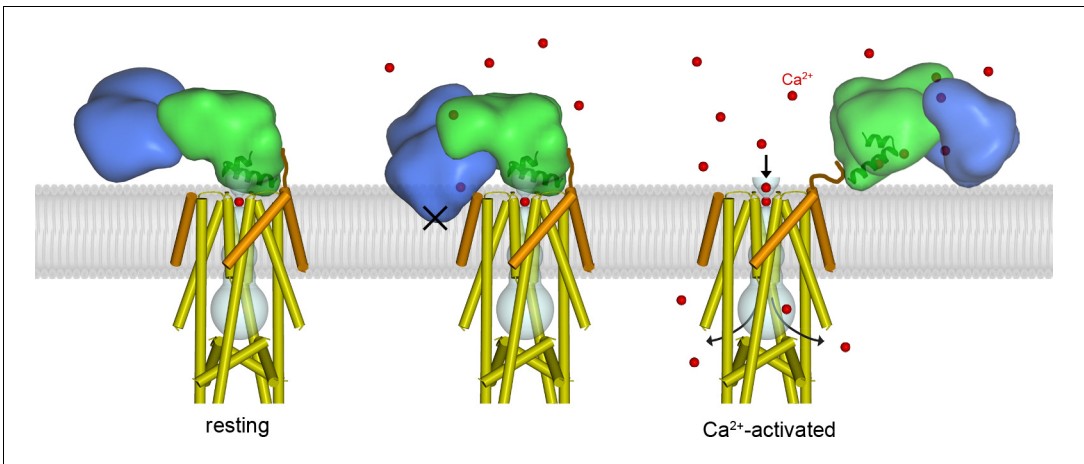

**Figure 5.** Proposed mechanism of Ca$^{2+}$-dependent control. Left, structure of the holocomplex under resting [Ca$^{2+}$]$_{IMS}$ conditions. MCU and EMRE are depicted with cylindrical helices; MICU1 and MICU2 are represented as semitransparent surfaces, with α1 and α2 helices of the UID as ribbons. The lipid membrane (gray) is based upon an atomistic model (*Figure 5—figure supplement 1*) . The UID blocks the pore (gray tube). Following elevation of [Ca$^{2+}$]$_{IMS}$, Ca$^{2+}$ binding causes the MICU1-MICU2 heterodimer to bend (the Ca$^{2+}$-bound MICU1-MICU2 structure is depicted with the channel in the center and right panels). The center panel indicates that bending would dislodge the UID from its receptor site in order to avoid thermodynamically unfavorable interactions of MICU2 with the membrane ('X'). The dislodged Ca$^{2+}$-bound MICU1-MICU2 heterodimer would no longer block the pore, thereby allowing Ca$^{2+}$ permeation through the channel, and it would be free to interact with the membrane (right). Upon the return of resting [Ca$^{2+}$], the heterodimer would resume its blocking conformation (left). One MICU1-MICU2 heterodimer is depicted and only one can bind to the receptor on the channel at a time but multiple MICU1-MICU2 heterodimers may be associated with the channel. A brown line depicts a hypothetical interaction of the acidic C-terminus of EMRE with MICU1.

The online version of this article includes the following figure supplement(s) for figure 5:

**Figure supplement 1.** Atomistic model of the resting holocomplex in a lipid bilayer.

**Figure supplement 2.** Model of a dimeric human MCU-EMRE-MICU1-MICU2 complex under resting [Ca$^{2+}$] conditions.

**Figure supplement 3.** Comparisons of the *Tc*MCU-EMRE-MICU1-MICU2 holocomplex structure with structures of human MCU-EMRE-MICU1-MICU2 holocomplexes.

MICU2 structure on the UID in the resting holocomplex reveals that the Ca$^{2+}$-induced conformational changes in the heterodimer would place hydrophilic regions of MICU2 within the membrane, which would be highly unfavorable thermodynamically (*Figure 5*, middle). Rather than inserting into the membrane, we hypothesize that the bending motion of MICU1-MICU2 acts like a lever against the membrane to pry MICU1 from the mouth of the pore when [Ca$^{2+}$]$_{IMS}$ is elevated. Additionally, the Ca$^{2+}$-dependent conformational changes within MICU1 cause a rotation of the UID that could disfavor binding to the channel (*Figure 4E*). The conformational changes would dislodge the UID from the blocking conformation, which would allow Ca$^{2+}$ ions to permeate though the channel while [Ca$^{2+}$]$_{IMS}$ remains elevated. A dislodged Ca$^{2+}$-bound MICU1-MICU2 complex would be free to bind to the surface of the inner mitochondrial membrane, with which it would associate due to its inherent affinity for membranes, and it may remain proximal the channel as a result of electrostatic interactions with the acidic tail of EMRE (*Figure 5*, right) (*Wang et al., 2019*). When resting [Ca$^{2+}$]$_{IMS}$ is restored, a MICU1-MICU2 complex, again devoid of its Ca$^{2+}$ ligands, would resume the blocking configuration (*Figure 5*, left, *Figure 5—figure supplement 1*). This mechanism of MICU1-MICU2 control is compatible with the observed dimerization of human MCU-EMRE channels that is due to association of their NTDs – there is sufficient room for MICU1-MICU2 complexes to associate with both channels of the dimer and to control Ca$^{2+}$ influx through their pores (*Figure 5—figure supplement 2*).

## Discussion

With analogy to classical toxin-channel interactions (*MacKinnon and Miller, 1988*), the structure of the holocomplex indicates that MICU1-MICU2 prevents mitochondrial $Ca^{2+}$ overload under resting conditions (*Liu et al., 2016*) and confers $Ca^{2+}$-dependent control by a pore-blocking mechanism. Operating like a regulatable toxin, resting levels of $Ca^{2+}$ permit the blocking conformation while elevated levels disrupt it. A single MICU1-MICU2 heterodimer appears capable of conferring these properties for the simple reason that the receptor site on the channel can accommodate only one UID. However, more than one MICU1-MICU2 heterodimer may be tethered to the channel at any given time, as is indicated by the cryo-EM analysis. Nucleation of more than one heterodimer in the vicinity of the channel would increase the local concentration of MICU1-MICU2 and facilitate a rapid response following restoration of resting cytosolic $Ca^{2+}$ levels. While MICU1-MICU2 visibly interacts with only one EMRE subunit, the displacement of an EMRE subunit (EMRE 4) from the TMD might serve as an additional mechanism to control ion flow since the expression of EMRE is necessary for $Ca^{2+}$ uptake by mitochondria (*Liu et al., 2016*; *Sancak et al., 2013*; *Tsai et al., 2016*). Additionally, the displacement of an EMRE subunit suggests that the number of EMRE subunits associated with an MCU tetramer can vary, as has been inferred from a recent study (*Payne et al., 2020*).

During the preparation of this manuscript, cryo-EM structures of a holocomplex comprising human MCU, EMRE, MICU1 and MICU2 subunits in resting and activated $[Ca^{2+}]$ conditions were reported (*Fan et al., 2020*). The conclusions of that work are congruous with the present study and validate the proposed mechanism of $Ca^{2+}$-dependent regulation of the uniporter by MICU1-MICU2. Unlike ours, which used lipid nanodiscs, the structures were determined using detergent-solubilized protein, but the absence of a lipid membrane-like environment does not seem to have dramatically influenced the binding of the MICU1-MICU2 regulatory complex to the channel under resting $Ca^{2+}$ conditions (*Figure 5—figure supplement 3*). Interestingly, an additional interaction with the 'D'-ring at the mouth of the pore, involving a loop region of MICU1 (amino acids 258–274) that is disordered in our structure, was observed (*Figure 5—figure supplement 3*). Weak density for this loop in our structure and its disorder in X-ray structures of isolated MICU1 proteins are indications of its flexibility (*Wang et al., 2014*). A structure of an intact holocomplex at elevated levels of $[Ca^{2+}]$, which was included in the recent work, indicates that the $Ca^{2+}$-bound MICU1-MICU2 regulatory complex binds adjacent to the pore in essentially the same manner that we propose (*Figure 5*, right, and *Figure 5—figure supplement 3*). The similarities of these studies, which used different experimental conditions, different protein constructs, and MCU-EMRE channels from different species, support a unified understanding of the mechanism of gatekeeping by MICU1-MICU2.

## Conclusion

This work reveals a mechanism for $Ca^{2+}$-dependent control of the mitochondrial calcium uniporter by the MICU1-MICU2 regulatory complex. The mechanism is distinct from the conformational changes that underlie gating, the process by which ion channels permit or prevent ion condution, in many other ion channels – these often involve changes in the dimensions of their ion conduction pores due to motions of transmembrane α-helices that line these pores, as was first exemplified by structural studies of potassium channels (*MacKinnon, 2003*). Our current understanding of how changes in $[Ca^{2+}]_{IMS}$ regulate the uniporter comes from the integration of insights from the present work and the large body of preceeding studies. The MICU1-MICU2 complex exists as a heterodimer that is located in the IMS. Under resting cytosolic $[Ca^{2+}]$ conditions, when the concentration of $Ca^{2+}$ is less than ~0.2 μM, $Ca^{2+}$-binding EF-hand domains within MICU1 and MICU2 are devoid of $Ca^{2+}$ and the heterodimer adopts a $Ca^{2+}$-free conformation that is relatively flat. An inherent affinity of MICU1-MICU2 for lipid membranes and an affinity for a flexible acidic region on the IMS portion of EMRE increase the local concentration of the heterodimer near the MCU/EMRE channel. MCU channels that have multiple EMRE subunits associated with them would tend to have more MICU1-MICU2 complexes nearby, as was inferred from a recent study (*Payne et al., 2020*). At these low resting concentrations of $Ca^{2+}$, a domain that is specific to MICU1, the uniporter interaction domain (UID), binds to a receptor site at the mouth of the channel's pore and blocks it, with analogy to how protein toxins from venomous organisms block neuronal cation channels. In this blocking conformation, the UID inhibits ion permeation that could otherwise dissipate the mitochondrial electromotive force. Although MICU2 interacts extensively with MICU1, it does not appear to contact the channel

directly. When the $Ca^{2+}$ concentration in the IMS becomes elevated, the binding of $Ca^{2+}$ to the EF-hand domains within MICU1 and MICU2 induce conformational changes in the MICU1-MICU2 complex that bend it and cause the UID to rotate. These conformational changes relieve blockage of the pore and thereby permit $Ca^{2+}$ ions to permeate through the channel while $[Ca^{2+}]_{IMS}$ remains elevated. These processes are reversed upon the restoration of resting $[Ca^{2+}]_{IMS}$ levels. MICU1-MICU2 heterodimers, and by analogy MICU1-MICU3 heterodimers in neurons, thus confer $Ca^{2+}$-dependent gating to the uniporter by operating like endogenous pore-blocking toxins, the binding of which is governed by $Ca^{2+}$-dependent conformational changes within them. This mechanism represents one of the ways that the mitochondrial $Ca^{2+}$ uniporter, an unusual and intricate ion channel complex, responds to physiological signals to regulate mitochondrial respiration and other cellular processes.

## Materials and methods

### Key resources table

| Reagent type (species) or resource | Designation | Source or reference | Identifiers | Additional information |
|---|---|---|---|---|
| Gene (*Tribolium castaneum*) | *Tc*MCU | IDT Inc | | |
| Gene (*Tribolium castaneum*) | *Tc*EMRE | IDT Inc | | |
| Gene (*Homo sapiens*) | *Hs*MICU1 | IDT Inc Genewiz, Inc. | | |
| Gene (*Homo sapiens*) | *Hs*MCIU2 | *IDT Inc* | | |
| Cell line (*H. sapiens*) | Expi293F | Sigma | A14527 | |
| Cell line (*H. sapiens*) | HEK-293T-MICU1-KO | *Kamer and Mootha, 2014* | DOI: 10.1002/embr.201337946 | |
| Chemical compound, drug | 1-palmitoyl-2-oleoyl-sn-glycero-3-phosphoethanolamine | Avanti Polar Lipids | 850757 | |
| Chemical compound, drug | 1-palmitoyl-2-oleoyl-sn-glycero-3-phospho-(1'-rac-glycerol) (sodium salt) | Avanti Polar Lipids | 840457 | |
| Chemical compound, drug | *1',3'-bis[1,2-dioleoyl-sn-glycero-3-phospho]-glycerol (sodium salt)* | Avanti Polar Lipids | 710335 | |
| Chemical compound, drug | Polyethylenimine, Linear, MW 25000, Transfection Grade (PEI 25K) | Polysciences, Inc | 23966–1 | |
| Chemical compound, drug | Sodium Butyrate | Sigma | 8451440100 | |
| Chemical compound, drug | n-Dodecyl-β-D-maltopyranoside | Anatrace | O310S | |
| Chemical compound, drug | (2α,3β,5α,15β,25R)—2,15-dihydroxyspirostan-3-yl O-β-D-glucopyranosyl-(1→3)-O-β-D-galactopyranosyl-(1→2)-O-[β-D-xylopyranosyl-(1→4)-β-D-galactopyranoside | Cayman Chemical Company | 14952 | |
| Chemical compound, drug | Membrane Scaffold Protein 1D1 | Sigma | M6574 | |
| Chemical compound, drug | 5-Methyl-2-oxo-4-imidazo lidinehexanoic acid | Sigma | D1411 | |
| Chemical compound, drug | Lipofectamine 3000 Transfection Reagent | Thermo Fisher | L3000008 | |
| Chemical compound, drug | *Calcium Green—5N, Hexapotassium Salt* | Thermo Fisher | C3737 | |

*Continued on next page*

*Continued*

| Reagent type (species) or resource | Designation | Source or reference | Identifiers | Additional information |
|---|---|---|---|---|
| Software, algorithm | MotionCor2 | *Zheng et al., 2017* | RRID:SCR_016499 | |
| Software, algorithm | CtfFind 4.1.10 | *Rohou and Grigorieff, 2015* | RRID:SCR_016731 | |
| Software, algorithm | RELION 3.0 | *Zivanov et al., 2018* | http://www2.mrc-lmb.cam.ac.uk/relion RRID:SCR_016274 | |
| Software, algorithm | SerialEM | *Glover, 2004* | RRID:SCR_017293 | |
| Software, algorithm | cryoSPARC v2 | Structura Biotechnology | https://cryosparc.com/ RRID:SCR_016501 | |
| Software, algorithm | PHENIX | *Adams et al., 2010* | https://www.phenix-online.org/ RRID:SCR_014224 | |
| Software, algorithm | COOT | *Emsley et al., 2010* | https://www2.mrc-lmb.cam.ac.uk/personal/pemsley/coot/ RRID:SCR_014222 | |
| Software, algorithm | PyMOL | *Schrödinger, 2020* | https://pymol.org/2/ RRID:SCR_000305 | |
| Software, algorithm | UCSF Chimera | *Pettersen et al., 2004* | https://www.cgl.ucsf.edu/chimera RRID:SCR_004097 | |
| Software, algorithm | GraphPad Prism 7 | GraphPad Software | https://cryosparc.com/ RRID:SCR_016501 | |
| Software, algorithm | Hole | *Smart et al., 1996* | http://www.holeprogram.org | |
| Software, algorithm | UCSF ChimeraX | Goddardnet al. 2018 | https://www.cgl.ucsf.edu/chimerax/ | |
| Others | QUANTIFOIL R1.2/1.3 holey carbon grids | Quantifoil | | |
| Others | FEI Vitrobot Mark IV | FEI Thermo Fisher | | |

## Holocomplex preparation

Tribolium castaneum MCU (*Tc*MCU; UniProt accession: TcasGA2_TC013837) and Tribolium castaneum EMRE (*Tc*EMRE; UniProt accession: TcasGA2_TC012057) were selected as candidates for structural and functional studies from among ~30 metazoan MCU orthologs that were evaluated using fluorescence-detection size-exclusion chromatography (FSEC) screening technique using HEK-293 cells (*Goehring et al., 2014*; *Wang et al., 2020*) (cells obtained from and validated by Invitrogen, tested negative for mycoplasma). cDNA encoding *Tc*MCU (residues 174–359, with a C-terminal Strep II tag) and *Tc*EMRE (residues 29–90) genes were chemically synthesized (IDT Inc), each cloned into a mammalian cell expression vector (*Goehring et al., 2014*) to encode proteins with an N-terminal Venus tag and an intervening PreScission protease cleavage site. The plasmids were co-transfected into Expi293 cells (obtained from and validated by Invitrogen, tested negative for mycoplasma) using PEI25k reagents (Polysciences, Inc) for transient expression. Briefly, 0.4 mg *Tc*MCU plasmid, 0.6 mg *Tc*EMRE plasmid and 3 mg PEI25k were mixed with 100 ml OptiMEM media (Invitrogen), incubated at room temperature for 20 min, and combined with approximately 3 $\times$ $10^9$ Expi293 cells in 1 L of Expi293 media (Invitrogen). After incubation of the cells at 37° C for 16 hr with shaking (125 rpm), 10 mM sodium butyrate (Sigma-Aldrich) was added, and the cells were cultured at 30° C for an additional 48 hr before harvest.

The pellet from 1 L of cell culture was resuspended in 100 ml lysis buffer (40 mM HEPES pH 7.5 200 mM NaCl, 0.15 mg/ml DNase I, 1.5 μg/ml Leupeptin, 1.5 μg/ml Pepstatin A, 1 mM AEBSF, 1 mM Benzamidine, 1 mM PMSF and 1:500 dilution of Aprotinin), and then solubilized by adding n-Dodecyl-β-D-Maltopyranoside (DDM, Anatrace) to a final concentration of 1% while stirring at 4°C for 1 hr. Solubilized proteins were separated from the insoluble fraction by centrifugation at 60,000 g for 1 hr and the

supernatant was filtered through a 0.22 µm polystyrene membrane (Millipore). 2 ml GFP nanobody resin was added and the sample was rocked at 4°C for 1 hr (*Kirchhofer et al., 2010*). The beads were washed with 100 ml of 20 mM HEPES pH7.5, 200 mM NaCl and 1 mM digitonin, eluted by adding PreScission protease (0.1 mg, 3 hr at 4°C, supplemented with 1 mM DTT), and further purified by size-exclusion chromatography (Superose 6 Increase, 10/300 GL column, GE Healthcare) equilibrated with 20 mM HEPES pH 7.5, 150 mM NaCl and 1 mM digitonin. The peak fractions were pooled, concentrated to $OD_{280}$ ~1.0 (100 kDa cutoff;~0.5 ml), mixed with 0.2 ml lipid/DDM mixture (17 mM DDM, 10 mM lipids: POPE (Avanti) : POPG (Avanti) : cardiolipin (18:1, Avanti) with a 2:2:1 wt ratio) and 0.25 ml nanodisc scaffold protein (MSP1D1, Sigma, 5 mg/ml, in buffer containing 20 mM Tris-HCl, pH 7.4, 100 mM NaCl, 0.5 mM EDTA and 5 mM sodium cholate). After 1 hr incubation at 4°C,~300 mg wet Bio-Beads SM2 (Bio-Rad) were added and the sample was rotated at 4°C for ~16 hr to remove detergent. To remove excess EMRE and empty nanodiscs, the sample was further purified by Strep-Tactin Sepharose (Qiagen). Briefly, the nanodisc sample was bound to 0.3 ml Strep-Tactin Sepharose with rotation at 4°C for 30 min, washed with 10 ml buffer (20 mM HEPES pH 7.5, 150 mM NaCl), and eluted with 2 ml buffer (20 mM HEPES pH 7.5, 150 mM NaCl and 5 mM d-Desthiobiotin).

MICU1 and MICU2 were expressed as a single polypeptide with a linker connecting the two proteins. The construct consists of human MICU1 (residues 94–476) connected to human MICU2 (residues 51–434). Unstructured regions of the polypeptide and a Ser-Asn peptide comprise a 44 amino acid linker between the structured regions of MICU1 and MICU2 (spanning ~32 Å in the structure). The expression construct was obtained in a stepwise manner. cDNA for human MICU1 (encoding residues 94–476) was amplified from a normal human brain cDNA library (BioChain, Inc) and ligated into the XhoI and EcoRI sites of an expression plasmid (*Goehring et al., 2014*) to contain an N-terminal Venus tag and PreScission protease cleavage site (Venus-PreScission-MICU1). cDNA encoding human MICU2 (amino acids 51–434) was amplified in the same manner and inserted into that plasmid, using MfeI/EcoRI and SalI restriction sites, to yield the final expression plasmid (Venus-PreScission-MICU1-MICU2). The expression and purification of MICU1-MICU2 followed a similar procedure to that for MCU and EMRE. The pellet from 0.3 L of cell culture (prepared using the same growth conditions as described above) was resuspended in 20 ml lysis buffer, DDM was added to a final concentration of 1%, and the sample was agitated at 4°C for 1 hr. The sample was clarified by centrifugation (60,000 g for 1 hr, 4°C) and the supernatant was filtered through a 0.22 µm polystyrene membrane (Millipore). 2 ml GFP nanobody resin was added and the sample was rocked at 4°C for 1 hr. The beads were washed with 100 ml buffer (20 mM HEPES pH 7.5, 500 mM NaCl and 1 mM DDM), the MICU1-MICU2 protein was eluted using 0.1 mg PreScission protease (16 hr, 4°C, supplemented with 1 mM DTT), and the sample was further purified by SEC (Superose 6 Increase, 10/300 GL) in 20 mM HEPES pH 7.5, 150 mM NaCl, 5 mM EDTA, 5 mM EGTA and 1 mM DDM. The peak fractions corresponding to the MICU1-MICU2 protein were pooled and concentrated to 1 mg/ml (Vivaspin 2, 100 kDa cutoff). To remove DDM,~300 mg wet Bio-Beads SM2 (Bio-Rad) were added and the sample was rotated (4°C, 3 hr). Purified MICU1-MICU2 protein was then combined with the purified with MCU-EMRE-nanodisc sample (using a molar ratio of 2 MICU1-MICU2 : one channel, 4°C, 30 min.) and the complex was purified by SEC (Superose 6 Increase, 10/300 GL column, GE Healthcare) in 20 mM HEPES pH 7.5, 150 mM NaCl, 5 mM EGTA. The peak fractions were collected, concentrated to 1 mg/ml (Vivaspin 2, 100 kDa cutoff), and used immediately for cryo-EM grid preparation. From the Maxchelator software (*Schoenmakers et al., 1992*), and assuming a typical trace concentration of $Ca^{2+}$ in buffer components (~2–5 µM), we estimate $[Ca^{2+}]_{free}$ ~100 pM in the sample.

## Preparation of $Ca^{2+}$-bound MICU1-MICU2 bound to nanodiscs

To form empty nanodiscs, 0.2 ml of a lipid/DDM mixture (17 mM DDM, 10 mM lipids: POPE: POPG: cardiolipin [18:1] with a 2:2:1 wt ratio) was combined with 0.25 ml nanodisc scaffold protein (MSP1D1, Sigma, 5 mg/ml). After 1 hr incubation on ice, 0.4 ml buffer (20 mM HEPES pH 7.5, 150 mM NaCl) and ~200 mg wet Bio-Beads SM2 (Bio-Rad) were added, the sample was rotated at 4°C for ~16 hr to remove detergent, and then it was purified by size-exclusion chromatography (Superose 6 Increase, 10/300 GL column, GE Healthcare; equilibrated with 20 mM HEPES pH 7.5, 150 mM NaCl). The peak fractions were pooled, combined with purified MICU1-MICU2 protein (using a molar ratio of 1 MICU1-MICU2 : 1.5 nanodisc scaffold protein; 4°C, 30 min.), and the complex was purified by size-exclusion chromatography (Superose 6 Increase, 10/300 GL column, GE Healthcare; equilibrated with 20 mM HEPES pH 7.5, 150 mM NaCl). The peak fractions were collected, concentrated

to 0.5 mg/ml (Vivaspin 2, 100 kDa cutoff), $CaCl_2$ was added to a final concentration of 1 mM, and the sample was used for cryo-EM grid preparation.

## EM sample preparation and data acquisition

For all samples, 4 µl of purified protein was applied to glow-discharged (10 s) Quantifoil R 1.2/1.3 grids (Au 400; Electron Microscopy Sciences) and plunge-frozen in liquid nitrogen-cooled liquid ethane, using a Vitrobot Mark IV (FEI) operated at 4°C with a blotting time of 2–3 s (blot force 0) with 100% humidity. Grids were clipped and loaded into a 300 keV Titan Krios microscope (FEI) equipped with a Gatan K3 direct electron detector. Micrographs were collected in super-resolution mode (pixel size of 0.532 Å) with a nominal defocus range of −1.0 to −3.0 µm. The dose rate was 20 electrons/pixel/s. Images were recorded for 4 s with 0.1 s subframes (40 total frames), corresponding to a total dose of 71 electrons per $Å^2$.

## Image processing and model building

For the holocomplex, 21,115 movie stacks were gain-corrected, twofold binned (using a pixel size of 1.1 Å), motion corrected, and dose weighted using MotionCor2 (*Zheng et al., 2017*). Contrast transfer function (CTF) estimates were performed in CTFFIND4 using non-dose weighted micrographs (*Rohou and Grigorieff, 2015*). 19,972 micrographs with CtfMaxResolution values better than 4 Å were selected for further processing. 17,440,131 particles were picked automatically and extracted (using a binned pixel size of 4.4 Å) using RELION 3.0 (*Zivanov et al., 2018*). These particles were then imported into cryoSPARC v.2 (*Punjani et al., 2017*). The particles were cleaned-up by one round of Ab initio reconstruction and two rounds of heterogeneous refinement. 2,388,041 particles from the best classes were selected and subjected to non-uniform refinement in cryoSPARC v.2, which yielded a reconstruction at ~9.4 Å overall resolution. The refined particles were then centered and re-extracted in RELION 3.0 with a pixel size of 2.2 Å, imported into cryoSPARC v.2 for another two rounds of heterogeneous refinement. 761,813 particles from the best classes were chosen and used for non-uniform refinement in cryoSPARC v.2, which yielded a reconstruction at ~4.4 Å overall resolution. The refined particles were centered and re-extracted in RELION 3.0 with a pixel size of 1.1 Å, and imported into cryoSPARC v.2 for another two rounds of heterogeneous refinement. 510,869 particles from the best class were subjected into non-uniform refinement in cryoSPARC v.2, and this yielded a reconstruction at 3.9 Å overall resolution. One round of Bayesian polishing in Relion 3.0 improved the resolution to 3.6 Å. After local CTF refinement in cryoSPARC v.2 and another round of Bayesian polishing in RELION 3.0, the particles were further classified by heterogeneous refinement in cryoSPARC v.2 and 3D classification (focused classification on the TMD) in RELION 3.0. 350,160 particles from the classes with a more well-defined TMD region were chosen and subjected to 3D auto refinement in RELION 3.0. Postprocessing in RELION 3.0 (using a calibrated pixel size of 1.064 Å) yielded the final reconstruction at 3.3 Å. All resolution estimates are based on gold-standard Fourier shell correlation (FSC) calculations. Estimation of the local resolution of the map was performed using Resmap (*Kucukelbir et al., 2014*).

For the structure of MICU1-MICU2 bound to nanodiscs and in 1 mM $Ca^{2+}$, 4675 movie stacks were gain-corrected, twofold binned (using a pixel size of 1.064 Å), motion corrected, and dose weighted using MotionCor2 (*Zheng et al., 2017*). Contrast transfer function (CTF) estimates were performed in CTFFIND4 using non-dose weighted micrographs (*Rohou and Grigorieff, 2015*). 4280 micrographs with CtfMaxResolution values better than 5 Å were selected for further processing. Particles were picked using Gaussian blob or 2D template-based autopicking and extracted (3,957,711 or 4,397,598 particles, respectively, with a binned pixel size of 2.128 Å) using RELION 3.0 (*Zivanov et al., 2018*). The two sets of particles were then imported into cryoSPARC v.2 to do processing separately (*Punjani et al., 2017*). For the particles from Gaussian autopicking, the particles were cleaned-up by one round of Ab initio reconstruction and three rounds of heterogeneous refinement (using seven classes), and 135,522 particles from the best classes were selected and subjected to non-uniform refinement in cryoSPARC v.2, which yielded a reconstruction at ~4.3 Å overall resolution. For the particles picked using 2D templates, particles were cleaned-up by three rounds of heterogeneous refinement and 152,512 particles from the best classes were selected and subjected to non-uniform refinement in cryoSPARC v.2, which yielded a reconstruction at ~4.3 Å overall resolution. The two sets of particles were then merged, centered and re-extracted in RELION 3.0 with a pixel size of 1.064 Å, and duplicate

particles or particles with CtfMaxResolution values worse than 4 Å were removed. The resulting 181,934 particles were imported into cryoSPARC v.2 for additional rounds of heterogeneous refinement. 142,055 particles from the best classes were chosen and used for non-uniform refinement in

**Table 1.** Data collection, refinement, and validation statistics.

| | Holocomplex at low [Ca²⁺] PDB: 6XQN EMDB: EMD-22290 | Ca²⁺-bound MICU1-MICU2 PDB: 6XQO EMDB: EMD-22291 |
|---|---|---|
| **Data collection and processing** | | |
| Microscope | FEI Titan Krios (at MSKCC) | FEI Titan Krios (at MSKCC) |
| Camera | Gatan K3 | Gatan K3 |
| Magnification | 22,500× | 22,500× |
| Voltage (kV) | 300 | 300 |
| Electron exposure (e⁻/Å²) | 71 | 71 |
| Defocus range (μm) | −1.0 ~ −3.0 | −0.8 ~ −2.3 |
| Pixel size (Å) | 1.064 (0.532)* | 1.064 (0.532)* |
| Software | RELION 3.0, cryoSPARC v2 | RELION 3.0, cryoSPARC v2 |
| Symmetry imposed | C1 | C1 |
| Initial particle images (no.) | 17,440,131 | 4,397,598 |
| Final particle images (no.) | 350,160 | 115,687 |
| Overall map resolution (Å) FSC threshold 0.143 | 3.3 | 3.1 |
| Local map resolution range (Å) | 3.0–5.0 | 2.8–5.0 |
| **Refinement** | | |
| Software | Phenix 1.13 real-space-refine | Phenix 1.13 real-space-refine |
| Initial model used (PDB code) | N/A | N/A |
| Model resolution (Å) FSC threshold 0.5 | 3.6 | 3.5 |
| Map sharpening $B$ factor (Å²) | −77 | −38 |
| **Model composition** | | |
| Non-hydrogen atoms | 9433 | 4225 |
| Protein residues | 1290 | 566 |
| Ligands | 1 (Calcium ion) | 4 (Calcium ions) |
| Water | 0 | 0 |
| **$B$ factors (Å²)** | | |
| Protein | 124.29 | 88.3 |
| Ligand | 70.1 | 94.9 |
| **R.m.s. deviations** | | |
| Bond lengths (Å) | 0.006 | 0.004 |
| Bond angles (°) | 1.027 | 0.569 |
| **Validation** | | |
| MolProbity score | 1.85 | 2.25 |
| Clashscore | 7.13 | 10.75 |
| **Ramachandran plot** | | |
| Favored (%) | 97.45 | 95.65 |
| Allowed (%) | 2.55 | 4.35 |
| Disallowed (%) | 0.0 | 0.0 |

*Super-resolution pixel size.

cryoSPARC v.2, which yielded a reconstruction at ~3.7 Å overall resolution. One round of Bayesian polishing in Relion 3.0 improved the resolution to 3.5 Å. After another round of Bayesian polishing in RELION 3.0, followed by CTF refinements (global and local) and several rounds of heterogeneous refinement in cryoSPARC v.2 (139,974 particles were selected), the resolution was improved to 3.2 Å. The selected particles were subjected to additional Bayesian polishing in RELION 3.0, followed by CTF refinement (global and local) and heterogeneous refinement in cryoSPARC v.2 (yielding 115,687 particles), which yielded the final reconstruction at 3.1 Å resolution. All resolution estimates are based on gold-standard Fourier shell correlation (FSC) calculations. Estimation of the local resolution of the map was performed using Resmap (*Kucukelbir et al., 2014*).

The atomic model of the holocomplex was manually built and refined in real space using the COOT software (*Emsley et al., 2010*). A cryo-EM structure of *Tc*MCU-EMRE (*Wang et al., 2020*) and X-ray structures of MICU1 and MICU2 (*Kamer et al., 2019*; *Wang et al., 2014*) were used as starting points. Further real-space refinement was carried out in PHENIX (*Adams et al., 2010*), to yield the final model (*Table 1*). Structural figures were prepared with Pymol (pymol.org) (*Schrödinger, 2020*), Chimera (*Pettersen et al., 2004*), ChimeraX (*Goddard et al., 2018*), and HOLE (*Smart et al., 1996*).

## Mitochondrial Ca$^{2+}$ uptake experiments

cDNA encoding full-length wild type human MICU1 was amplified from a normal human brain cDNA library (BioChain, Inc), whereas an EF-hand mutant MICU1 gene with mutations abolishing Ca$^{2+}$ binding (D231A, E244K, D421A, E432K) was chemically synthesized (Genewiz, Inc). Both genes were sub-cloned into the XhoI and EcoRI sites of a mammalian expression vector that includes a C-terminal Rho-1D4 antibody tag (*Baradaran et al., 2018*; *Molday and MacKenzie, 1983*). Mutations F106A, R117A, Y121A, K126A and R129A were generated by PCR using mutagenic primers on the background of the EF-hand mutant and verified by sequencing. MICU1 knockout cells, derived from human HEK-293T cells and generously provided by V. Mootha (*Kamer and Mootha, 2014*) (authenticated using mitochondrial Ca$^{2+}$ uptake assays (*Figure 3—figure supplement 5*); tested negative for mycoplasma), were grown in DMEM media (Invitrogen). 3 μg of each plasmid was transfected into ~$1.5 \times 10^6$ of these cells using Lipofectamine 3000 (Invitrogen), and the cells were grown for an additional 24 hr at 37°C before use.

The mitochondrial Ca$^{2+}$ uptake assay using these cells was performed as described (*Baradaran et al., 2018*), with slight modifications. Briefly, after addition of 1 μM Calcium Green-5N (Life Technologies) and 0.05 mM digitonin, 20 μM CaCl$_2$ (from a 4 mM stock in water) was added at the 100 s time point to initiate mitochondrial Ca$^{2+}$ uptake. The data plotted in *Figure 3E* were obtained from Ca$^{2+}$ uptake curves (e.g. *Figure 3—figure supplement 5*) in the following manner. A rate of Ca$^{2+}$ uptake (uptake_rate) was defined as the slope of a linear fit of the Ca$^{2+}$ uptake curve between 111 and 135 s. Relative uptake was then defined as: (uptake_rate$_{UID-mutant}$ − uptake_rate$_{EF-hand-mutant}$) / (uptake_rate$_{WT}$ − uptake_rate$_{EF-hand-mutant}$), where 'WT' represents wild type MICU1, 'EF-hand-mutant' represents MICU1 bearing the EF-hand mutations, and 'UID-mutant' represents a given mutation of the UID on the background of the EF-hand-mutant. From this equation, a relative uptake of 1 indicates a mitochondrial Ca$^{2+}$ uptake rate equivalent to that observed for wild type MICU1 and a relative uptake of 0 indicates a mitochondrial Ca$^{2+}$ uptake rate equivalent to that observed for the EF-hand-mutant.

## Assessing the association of MICU1-MICU2 with liposomes

To prepare liposomes, 5 mg of dried lipids (POPE: POPG: cardiolipin [18:1] with a 2:2:1 wt ratio) was mixed with 1 ml reconstitution buffer (20 mM HEPES pH 7.5, 150 mM NaCl) and the sample was sonicated until homogeneous (~2 min). 20 μl of purified MICU1-MICU2 protein (0.5 mg/ml) was added to 80 μl of liposomes or 80 μl of reconstitution buffer, each supplemented with a final concentration of 1 mM CaCl$_2$ or with 1 mM EGTA and 1 mM EDTA. As a negative control, 20 μg of purified green fluorescent protein was combined with liposomes or buffer in the same manner. After incubation for 30 min at 4°C, the samples were centrifuged at 140,000 g (30 min at 4°C). The supernatants were removed and the pellets were resuspended for SDS-PAGE analysis (by resuspending the pellets in 40 μl reconstitution buffer, adding 40 μl SDS-PAGE loading buffer containing 100 mM DTT, and analyzing 20 μl by Coomassie-stained SDS-PAGE).

## Materials availability

## Acknowledgements

We thank RK Hite, CD Lima, members of the Long laboratory (TL Benz, X Hou and Y Jiang), and N Paknejad for discussions. We thank V Mootha for the MICU1 knockout cell line and MJ de la Cruz of the Memorial Sloan Kettering Cancer Center Cryo-EM facility for help with data collection. We thank J Goldberg for spectrophotometer use.

## Additional information

### Funding

| Funder | Grant reference number | Author |
| --- | --- | --- |
| National Institute of General Medical Sciences | R35GM131921 | Stephen Barstow Long |
| National Cancer Institute | P30CA008748 | Stephen Barstow Long |
| National Institute of General Medical Sciences | 5T32GM008539 | Bryce D Delgado |

The funders had no role in study design, data collection and interpretation, or the decision to submit the work for publication.

### Author contributions

Chongyuan Wang, Conceptualization, Data curation, Formal analysis, Validation, Investigation, Visualization, Methodology, Writing - original draft, Writing - review and editing; Agata Jacewicz, Conceptualization, Data curation, Formal analysis, Investigation, Writing - original draft, Writing - review and editing; Bryce D Delgado, Data curation, Investigation, Writing - review and editing; Rozbeh Baradaran, Conceptualization, Data curation, Formal analysis, Investigation; Stephen Barstow Long, Conceptualization, Resources, Formal analysis, Supervision, Funding acquisition, Validation, Visualization, Methodology, Writing - original draft, Project administration, Writing - review and editing

### Author ORCIDs

Stephen Barstow Long https://orcid.org/0000-0002-8144-1398

### Decision letter and Author response

Decision letter https://doi.org/10.7554/eLife.59991.sa1
Author response https://doi.org/10.7554/eLife.59991.sa2

## Additional files

### Supplementary files

• Transparent reporting form

### Data availability

The atomic coordinates and EM maps have been deposited in the Protein Data Bank (www.rcsb.org) and the EMDB (www.ebi.ac.uk/pdbe/emdb/): PDB ID 6XQN, EMDB ID EMD-22290 (holocomplex in low Ca2+); PDB ID 6XQO, EMDB ID EMD-22291 (Ca2+-bound MICU1-MICU2 heterodimer).

The following datasets were generated:

| Author(s) | Year | Dataset title | Dataset URL | Database and Identifier |
| --- | --- | --- | --- | --- |
| Wang C, Jacewicz A, Delgado BD, | 2020 | holocomplex in low Ca2+ | http://www.ebi.ac.uk/pdbe/entry/emdb/EMD- | Electron Microscopy Data Bank, EMD- |

| | | | | |
|---|---|---|---|---|
| Baradaran R, Long SB | | | 22290 | 22290 |
| Wang C, Jacewicz A, Delgado BD, Baradaran R, Long SB | 2020 | holocomplex in low Ca2+ | http://www.rcsb.org/structure/6XQN | RCSB Protein Data Bank, 6XQN |
| Wang C, Jacewicz A, Delgado BD, Baradaran R, Long SB | 2020 | Ca2+-bound MICU1-MICU2 heterodimer | http://www.ebi.ac.uk/pdbe/entry/emdb/EMD-22291 | Electron Microscopy Data Bank, EMD-22291 |
| Wang C, Jacewicz A, Delgado BD, Baradaran R, Long SB | 2020 | Ca2+-bound MICU1-MICU2 heterodimer | http://www.rcsb.org/structure/6XQO | RCSB Protein Data Bank, 6XQO |

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
