## [Decision Letter]

**Acceptance summary:**

This manuscript reports cryo-EM structures of the mitochondrial calcium uniporter (MCU) in complex with the auxiliary subunit EMRE and the MICU1/MICU2 regulatory heterodimer. The authors also present structures of Ca^2+^-bound MICU1/MICU2 interacting with nanodiscs. Together, these structures provide remarkable insight into the mechanism by which the MICU1/MICU2 heterodimer gates the pore, occluding the mouth of the pore and preventing flux under resting (low) calcium conditions. The authors draw an analogy to pore-blocking toxins of potassium and sodium channels that inhibit by physically occluding the ion permeation pathways of these channels. The authors suggest a mechanism by which a conformational change in MICU1/MICU2 upon Ca^2+^ binding disrupts the binding to the MCU pore, displacing it to the membrane to support ion conduction. This is a very strong and impressive manuscript. The analysis of the structures is rigorous and well-supported by the literature and the transport data in the manuscript. The work is of high quality, expands and strongly enhances the results for human holocomplex recently published in Nature (Fan et al., 2020). The manuscript itself is clearly written and we think it should be published as soon as possible.

**Decision letter after peer review:**

Thank you for submitting your article "Structures reveal gatekeeping of the mitochondrial Ca^2+^ uniporter by MICU1-MICU2" for consideration by *eLife*. Your article has been reviewed by three peer reviewers, and the evaluation has been overseen by Kenton Swartz as the Reviewing Editor and Senior Editor. The following individual involved in review of your submission has agreed to reveal their identity: Alexander Sobolevsky (Reviewer #3).

The reviewers have discussed the reviews with one another and the Reviewing Editor has drafted this decision to help you prepare a revised submission.

Summary:

This manuscript from Stephen Long's lab presents structures of the mitochondrial calcium uniporter (MCU) in complex with the auxiliary subunit EMRE and the MICU1/MICU2 regulatory heterodimer. The authors also present structures of Ca^2+^-bound MICU1/MICU2 interacting with nanodiscs. Together, these structures provide insight into the mechanism by which the MICU1/MICU2 heterodimer gates the pore, occluding the mouth of the pore and preventing flux under resting (low) calcium conditions. The authors draw an analogy to sodium channel inhibition by the pore-binding conotoxin KIIIA. The authors suggest a mechanism by which a conformational change in MICU1/MICU2 upon Ca^2+^ binding disrupts the binding to the MCU pore, displacing it to the membrane. This is a very strong and impressive manuscript. The analysis of the structures is rigorous and well-supported by the literature and the transport data in the manuscript. The work is of high quality, expands and strongly enhances the results for human holocomplex recently published in Nature (Fan et al., 2020). The manuscript itself is clearly written and we think it should be published as soon as possible.

Essential revisions:

1) The manuscript did not comment on the function of the beetle MCU(δ NTD)-EMRE channel core and whether it showed MICU1-dependent Ca^2+^ regulation. Would it be possible to show that the construct used was actually functional?

2) The MICU1-MICU2 concatemer consists of human MICU1 (residues 94-476) connected to MICU2 (residues 51-434) by a 44 amino acid linker between the structured regions of MICU1 and MICU2. Considerable portions of both MICU1 and MICU2 were removed from the construct and an artificial linker was introduced to connect MICU1 and MICU2. Do the authors have data showing that this concatemer actually functions like the native MICU1-MICU2 heterodimer.

3) The authors suggested that the more bent MICU1-MICU2-Ca^2+^ acts like a lever against the membrane to pry MICU1 from the mouth of the pore. However, Figure 5 (right panel) also indicates that the MICU1-MICU2 dimer would need to rotate 180 degrees to unblock the channel. Please consider revising Figure 5 and also Figure 5—figure supplement 3. Another point regarding the lever mechanism. If MCU-EMRE resides in a curved membrane as indicated by the structure of the channel dimer (Figure 5—figure supplement 3), the bent MICU1-MICU2-Ca^2+^ may not be able to force MICU1 out of the mouth of the pore. Alternative/additional discussion should be added to address this possibility.

4) In high calcium conditions, MICU1-MICU2 heterodimers form dimers through MICU2-MICU2 interface as part of the recently published O-shaped uniplex (Fan et al., 2020). Is this dimer of heterodimers formation observed (may be in a small subset of particles) for the calcium-bound MICU1-MICU2 heterodimer in the present study?

5) Is it possible to map the second MICU1-MICU2 heterodimer binding site on the MCU-EMRE-MICU1-MICU2 holocomplex observed in 2% of channel assemblies? Even at low resolution, it might be interesting to analyze this second binding interface. Would this interface be the same/different/overlap with the putative interface between MICU1 and MCU-EMRE in the O-shaped uniplex? Depending on the answer, it might be possible to speculate about the physiological relevance of this interface.

---

## [Author Response]

Essential revisions:1) The manuscript did not comment on the function of the beetle MCU(δ NTD)-EMRE channel core and whether it showed MICU1-dependent Ca^2+^ regulation. Would it be possible to show that the construct used was actually functional?

Thank you for these comments. As mentioned in the revised manuscript, we shown in a recent study from our laboratory (Wang, Baradaran and Long, 2020) that the MCU/EMRE constructs used for structural studies catalyze Ca^2+^ uptake into human mitochondria (Supplementary Figure 1B of that manuscript, trace labeled “*Tc*MCU_ΔNTD_ and EMRE coexpression"). Preliminary functional studies from our laboratory suggest that human MICU1-MICU2 confers regulation to *Tc*MCU/EMRE in a normal manner; the data will be published when they are complete. Several other lines of evidence support this conclusion. Ming-Feng Tsai’s laboratory has shown that human MICU1 can bind to MCU/EMRE channels from numerous metazoan species (Phillips, Tsaiand Tsai, 2019). Structural comparisons show that the UID receptor surface on the TcMCU/EMRE channel is indistinguishable from that of human MCU/EMRE (Figure 1—figure supplement 6). Sequence alignments indicate that the residues of human MICU1 that interact with TcMCU/EMRE in the structure are identically conserved in beetle MICU1 (Figure 3—figure supplement 2). Finally, the transport data presented in our manuscript support the conclusion that human MICU1-MICU2 regulatory complexes bind to human MCU/EMRE channels in the manner observed.

2) The MICU1-MICU2 concatemer consists of human MICU1 (residues 94-476) connected to MICU2 (residues 51-434) by a 44 amino acid linker between the structured regions of MICU1 and MICU2. Considerable portions of both MICU1 and MICU2 were removed from the construct and an artificial linker was introduced to connect MICU1 and MICU2. Do the authors have data showing that this concatemer actually functions like the native MICU1-MICU2 heterodimer.

Thank you for this comment. Preliminary data from both mitochondrial Ca^2+^ uptake studies and in vitro reconstitution experiments, as mentioned in response to Comment 1, suggest that the MICU1-MICU2 concatemer functions like a native MICU1-MICU2 heterodimer. In support of this, we show that the MICU1-MICU2 concatemer adopts the same structural conformation as other available structures of MICU1/MICU2 complexes (Figure 4—figure supplement 6). The effects of mutations within UID of human MICU1 on mitochondrial Ca^2+^ uptake rates (Figure 3E) support the conclusion that native MICU1-MICU2 heterodimer complexes bind to MCU/EMRE channels in the manner observed.

3) The authors suggested that the more bent MICU1-MICU2-Ca^2+^ acts like a lever against the membrane to pry MICU1 from the mouth of the pore. However, Figure 5 (right panel) also indicates that the MICU1-MICU2 dimer would need to rotate 180 degrees to unblock the channel. Please consider revising Figure 5 and also Figure 5—figure supplement 3. Another point regarding the lever mechanism. If MCU-EMRE resides in a curved membrane as indicated by the structure of the channel dimer (Figure 5—figure supplement 3), the bent MICU1-MICU2-Ca^2+^ may not be able to force MICU1 out of the mouth of the pore. Alternative/additional discussion should be added to address this possibility.

Thank you for these comments regarding the proposed mechanism. With regard to the right panel of Figure 5, we do not intend to imply that the “MICU1-MICU2 dimer would need to rotate 180 degrees to unblock the channel”. Bending of the MICU1-MICU2 dimer itself would relieve block, and we have attempted to clarify our thinking in this regard. However, because Ca^2+^-bound MICU1-MICU2 associates with lipid membranes (as evidenced by our studies using nanodiscs and liposomes), we hypothesize that the dimer would usually be associated with the surface of the inner mitochondrial membrane when Ca^2+^ levels are elevated. Due to tethering to EMRE, we propose that it would typically bind to the region of the membrane adjacent to the channel in a manner similar to what is shown in the right panel of Figure 5. In the revised manuscript we also point out that Ca^2+^-binding causes the UID to rotate with respect to the remainder of MICU1 (Figure 4E) and that this might also reduce its affinity for the channel. With respect to the situation in which channel dimers may reside in a curved membrane: We still posit that the bent MICU1-MICU2-Ca^2+^ would be able to force MICU1 out of the mouth of the pore because the membrane would need to be considerably more bent than it is in the channel dimer complex to prevent this effect. However, it is conceivable that membrane curvature could modulate the affinity of the UID for its receptor, and one could imagine that this may modulate Ca^2+^ uptake by the channel in regions of high curvature (e.g. cristae). In the revised manuscript, we draw more attention to the Ca^2+^-induced rotation of the UID as an additional means by which the affinity of the UID may be reduced when [Ca^2+^}_IMS_ is elevated.

4) In high calcium conditions, MICU1-MICU2 heterodimers form dimers through MICU2-MICU2 interface as part of the recently published O-shaped uniplex (Fan et al., 2020). Is this dimer of heterodimers formation observed (may be in a small subset of particles) for the calcium-bound MICU1-MICU2 heterodimer in the present study?

2D classification of the Ca^2+^-bound MICU1-MICU2 dataset reveals approximately 3% of particles containing two MICU1-MICU2 complexes associated with each other (Figure 4—figure supplement 1B). These particles display an orientation preference such that “top views” predominate. This orientation preference prevented us from generating an accurate 3D reconstruction. However, these 2D class averages are strikingly similar to projections of the dimer of MICU1-MICU2 heterodimers from the high [Ca^2+^] form of the human holocomplex structure (Figure 4—figure supplement 1C). This similarity suggests that a portion of the MICU1-MICU2 heterodimers do form dimers through MICU2:MICU2 interfaces in our cryo-EM sample. However, because MICU1 and MICU2 have similar structures, we cannot exclude the possibility that some fraction of MICU1-MICU2 heterodimers form dimers through MICU1:MICU1 interfaces in our sample as well.

5) Is it possible to map the second MICU1-MICU2 heterodimer binding site on the MCU-EMRE-MICU1-MICU2 holocomplex observed in 2% of channel assemblies? Even at low resolution, it might be interesting to analyze this second binding interface. Would this interface be the same/different/overlap with the putative interface between MICU1 and MCU-EMRE in the O-shaped uniplex? Depending on the answer, it might be possible to speculate about the physiological relevance of this interface.

Thank you for these comments. We have generated a 3D reconstruction from the ~2% of particles of the holocomplex dataset at low Ca^2+^ that contains two MICU1-MICU2 heterodimers per channel assembly (Figure 1—figure supplement 3C). As described in the revised manuscript, although this structure is determined at low (~10 Å resolution), key features are visible. One MICU1-MICU2 heterodimer occupies the site observed in the high-resolution holocomplex structure and its association with the MCU/EMRE channel is indistinguishable from that structure. Density for the other MICU1-MICU2 heterodimer, which is considerably weaker, is positioned to the side, next to channel. For clarity, we note that, to our knowledge, a structure of an “O-shaped uniplex” that contains two channels and two MICU1-MICU2 heterodimers has not been determined under resting (low) levels of Ca^2+^. However, a speculative model of such a complex is shown in Figure 5—figure supplement 2. The second MICU1-MICU2 heterodimer within the low-resolution structure could potentially occupy a similar location as the second MICU1-MICU2 heterodimer in the O-shaped uniplex model. However, we prefer to not speculate on the potential physiological relevance of this, partly because we have shown that MICU1-MICU2 complexes have an affinity for membranes and this may be the reason for an additional MICU1-MICU2 complex bound in a small percentage of the particle images.